# Exploring positive experiences of primary and secondary caregivers of older persons in resource-limited urban settings in Accra, Ghana

**Frank Kyei-Arthur** [1]* , **Samuel Nii Ardey Codjoe**[2] , **Delali Margaret Badasu**[2]

**1** Department of Environment and Public Health, University of Environment and Sustainable Development, Somanya, Eastern Region, Ghana, **2** Regional Institute for Population Studies, University of Ghana, Legon, Accra, Ghana

☯ These authors contributed equally to this work.

\* fkyeiarthur@yahoo.com

**Data Availability Statement:** All relevant data are within the paper and its Supporting Information files.

## Abstract

Family caregivers experience both negative and positive outcomes. However, most studies have mainly focused on the negative outcomes. In addition, few studies have focused on both primary and secondary caregivers. This study explored the positive experiences of primary and secondary caregivers of older persons in resource-limited urban settings in Accra, Ghana. This study used phenomenological design, and thirty-one family caregivers were interviewed in James Town and Ussher Town. The interviews were audio-recorded, transcribed verbatim, and analysed thematically using NVivo 10. The findings show that the primary and secondary caregivers derived tangible and intangible rewards from providing care to older persons. The tangible rewards included gifts, while the intangible include blessings, skills acquisition, enhanced personal attributes, family cohesion, and health consciousness. Positive caregiving experiences can mitigate caregivers' burden and burnout. Therefore, policymakers and social workers should design interventions that will enhance the positive experiences of family caregivers. They should also consider the gender and age of caregivers in designing these interventions.

## Introduction

Globally, there has been an increase in the proportion of older persons (aged 65 years and older). Between 1990 and 2021, the proportion of the population of older people to the total global population increased from 6 percent to 10 percent, and it is estimated to reach 16 percent in 2050 [1, 2]. In Ghana, the proportion has increased from 3 percent in 1960 to 4 percent in 2021 [2, 3]. By 2050, it is estimated that about 9 percent of Ghanaians will be aged 65 years and older. The increase in the proportion of older persons can be attributed to increasing longevity and declining fertility [4].

Older persons, especially those aged 75 years and over, may need some assistance due to their functional limitations. In Ghana and other sub-Saharan African countries, the family is

**Funding:** The author(s) received no specific funding for this work.

**Competing interests:** The authors have declared that no competing interests exist.

the main provider of care and support to older persons [3, 5, 6]. The provision of unpaid care to a relative, partner, friend, or neighbour who needs care is referred to as family caregiving [7]. Family caregiving roles are often performed by women [8–10]. In addition, women often provide personal care (such as bathing, cooking, laundry, and dressing). In contract, men provide financial care (such as giving money for food and paying medical and other types of bills) [9, 11].

In Ghana, modernisation processes (such as urbanisation) have influenced the role of the family in providing care to older persons due to financial constraints and increased involvement of women in the paid labour market [12–14]. For instance, the increased participation of women in the labour market affects the availability of women to provide personal care to older persons [15]. In addition, research has established that financial constraints limit the financial care provided to parents [16]. Also, poverty in coastal urban areas (including the study area, Ga Mashie) has worsened over time [17], and this has repercussions for financial care to older persons.

Previous studies have established that family caregivers experience both negative and positive outcomes from performing their roles and responsibilities [18–21]. For instance, researchers have shown that family caregiving enhances caregiver-recipient relationship [20–22] and caregivers' confidence [21]. Also, caregivers learn to handle difficult situations [21, 22] and derive satisfaction from their caregiving responsibilities [21, 23]. However, most studies on family caregiving to older persons have mainly focused on the negative outcomes to the neglect of the positive outcomes [21, 23, 24]. In this regard, previous studies have established that family caregivers of older persons experience financial constraints, poor physical and mental health, and weight loss [9, 13, 25, 26].

In addition, family caregivers often share their roles and responsibilities with others [27, 28]. In every caregiving situation, there is always someone who provides the most care. This person is referred to as the primary caregiver, while others who support this main person are known as secondary caregivers. However, most studies on family caregiving to older persons have mainly focused on primary caregivers [29–33], with limited studies on both primary and secondary caregivers [13, 26, 34]. The main focus on primary caregivers is partly because they provide most care and may experience a greater brunt of the negative outcomes of family caregiving. Nevertheless, family caregiving is dynamic, and caregiving roles change over time [21].

The available literature does not indicate that any study has explored the positive experiences of both primary and secondary caregivers of older persons in Ghana and sub-Saharan Africa in general. The paucity of studies on the positive experiences of primary and secondary caregivers in Ghana and sub-Saharan Africa limits researchers' and policymakers' understanding of the nuances of family caregiving experiences which would enable them to design appropriate interventions to address the associated challenges. Accordingly, this study explored the positive experiences of primary and secondary caregivers in urban resource-limited communities in Accra, the capital city of Ghana. It specifically explored the rewards that primary and secondary caregivers derived from providing care to older persons.

## Conceptual framework

This study was guided by Kramer's model of caregiver adaptation [35]. She highlights that well-being outcomes (whether positive or negative) are influenced by the background and context of caregiving, and intervening processes. According to Kramer [35], the background and context of caregiving component consist of caregiver characteristics (namely age, gender, duration of caregiving activities, and employment status), care recipient characteristics (that is, the type of illness, the severity of illness, care recipient needs), and caregiver attitudinal

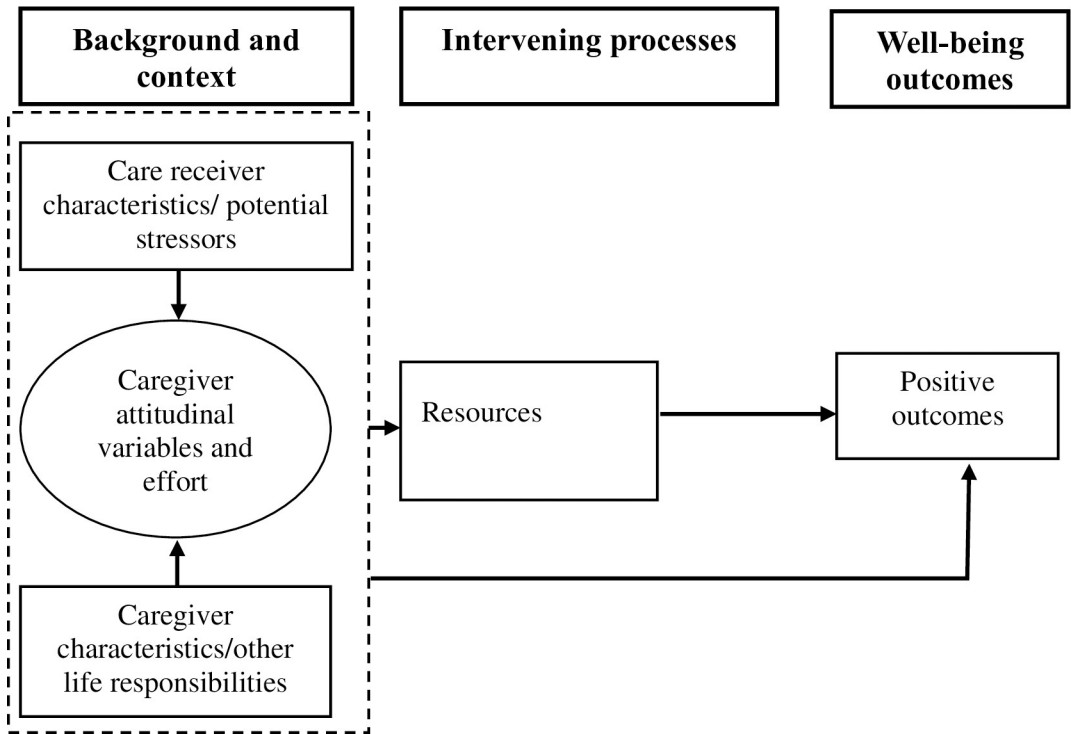

**Fig 1. Conceptual framework for positive experiences of primary and secondary caregivers.**

variables and efforts (such as motivations for providing care). In addition, resources (such as social support and coping strategies) and appraisal of role gain and strain in the form of caregiving satisfaction and burden were identified as factors that enhance and reduce positive and negative outcomes of caregivers, respectively.

Although Kramer's model of caregiver adaptation discusses both positive and negative outcomes, this study focused on the positive outcomes of caregivers. Therefore, it adopted and modified Kramer's model of caregiver adaptation (Fig 1). Positive outcomes were conceptualised as the rewards caregivers derive from providing care to their care recipients.

This manuscript is derived from a broader thesis that collected data on characteristics of family caregivers and their care recipients, and caregivers' motivations for providing care to their care recipients (Kyei-Arthur [Unpublished]). In addition, data were collected on family caregivers' social support and coping strategies. Previous studies have documented that background and context of caregiving, and intervening processes influence caregiver well-being outcomes [36–38]. Therefore, we also conceptualised that the background and context of caregiving can directly influence positive outcomes.

This manuscript does not seek to assess the pathways to positive caregiving, but it seeks to describe positive outcomes among family caregivers in a context that has rarely been studied.

## Methods

### Design and study setting

This study used phenomenology to explore the positive caregiving experiences of primary and secondary caregivers. Phenomenology describes people's lived experiences and how they

interpret their experiences [39, 40]. Phenomenology was selected because it assists investigators to explain the meaning people assign to their experiences [41, 42].

This study was conducted in James Town and Ussher Town, which are neighbouring communities in the city of Accra, commonly referred to as Ga Mashie. Ga Mashie is one of the six indigenous Ga localities and one of the oldest localities in Accra, the capital city of Ghana [43]. They are located on the Gulf of Guinea and recognised as resource-limited urban localities in Accra [44]. They are densely populated, and residents lack access to basic social amenities such as potable water and waste disposal. Fishing and fishing-related activities are the main economic activities of residents. Residents in James Town and Ussher Town dwell in multigenerational households.

### Data collection

The data collection for this study followed a two-stage procedure. In the first stage, the first author obtained the list and identifiers (e.g., structure, household and individual identifiers) of family caregivers who participated in the third wave of the Regional Institute for Population Studies' (RIPS) Survey dubbed "Urban Health and Poverty Survey" from the Principal Investigators. The first author was a research assistant during the third wave of the RIPS Survey, while the second and third authors were co-investigators on the RIPS Survey. The Urban Health and Poverty Survey took place in September 2013. It is a longitudinal survey that examines the health, poverty, and development nexus in Agbogbloshie, James Town, and Ussher Town. Overall, 57 family caregivers were involved in the third wave of the Urban Health and Poverty Survey.

In the second stage, the first author and trained research assistants contacted family caregivers involved in the RIPS Survey and they helped the research team (first author and trained research assistants) reach out to their care recipients. In total, the first author and trained research assistants were able to contact 30 out of the 57 family caregivers using the structure and household identifiers (e.g., enumeration area number, structure number of household, name of the head of household, and contact of the head of household) and family caregiver identifiers (e.g., name, and contact numbers). In addition, care recipients assisted the research team in reaching out to all persons who provided care to them since the study was interested in primary and secondary caregivers.

Generally, a family caregiver refers to a relative and non-relative (including friends) who provide unpaid care to persons in need of care. In this study, a primary caregiver refers to a family caregiver who performs most caregiving activities for their care recipient. In contrast, a secondary caregiver is a family caregiver who supports a primary caregiver to provide care to their care recipient. Secondary caregivers complement the care provided by primary caregivers.

Family caregivers who met the following criteria were interviewed: they must be 18 years and over; a relative or non-relative of the care recipient; currently providing the unpaid care to an older person; and should have provided unpaid care for at least six months prior to the data collection. Also, family caregivers who met the following criteria were excluded from the study: care recipients had died more than six months preceding the data collection; family caregivers being less than 18 years old; providing care to an older person less than six months preceding the data collection; and not willing to share their caregiving experiences.

The data collection for the study was conducted in November 2016. In total, thirty-one family caregivers were interviewed. The in-depth interviews were conducted in the *Ga* language by trained research assistants. The interviews ranged between 30 and 55 minutes. Ethical clearance for the study was received by the Ethics Committee for the Humanities (ECH) of the University of Ghana (ECH 009/16-17). Participation in the study was voluntary, and informed consent of caregivers was sought before they were interviewed.

## Data analysis

All the interviews were audio-taped and transcribed verbatim from the *Ga* language into English. The interview transcripts were quality checked and analysed using the thematic analysis approach [45]. All the transcripts were read several times to ensure a comprehensive understanding of the data. Codes were then assigned to important text in the form of categories and subcategories using NVivo 10 software. These categories and subcategories were then aggregated to develop themes of positive caregiving experiences.

Trustworthiness of findings is essential in qualitative research. To achieve trustworthiness in qualitative studies, numerous strategies have been recommended, including thick description, peer review/debriefing, development of a coding system, and member checking [46]. To achieve trustworthiness in the study, the researchers worked collaboratively and closely. Also, they consulted peers during the entire data analysis process to ensure that data were analysed from the perspectives of family caregivers. Their feedback on quotes and/or themes were addressed by re-analysing/re-examining those quotes and/or themes. Also, the study's findings after data analysis were shared with family caregivers to confirm if the analysis reflected their experiences or provide additional information. Eight family caregivers accepted to go through the analysis and confirmed the analysis reflected their experiences. They, however, did not provide any additional information.

## Results

### Socio-demographic characteristics of the family caregivers and their care recipients

Table 1 presents the socio-demographic characteristics of family caregivers. Family caregivers were aged between 21 and 76 years. Out of the 31 family caregivers, 15 were primary caregivers, while the remaining 16 were secondary caregivers. Most primary and secondary caregivers were females (73.3%) and Ga-Dangme (86.7%). The majority of the secondary caregivers (100%) had formal education than primary caregivers (93.3%). Most primary and secondary caregivers (77.4%) were employed, and more than half (75%) worked at home or close to home. In addition, 27 of the respondents were Christians, and 20 had ever married.

Furthermore, most of the caregivers (71%) lived in James Town. Nine of the primary caregivers and eight (8) of the secondary caregivers were sons/daughters and other relatives of the care recipients, respectively.

In terms of caregiving experience, primary caregivers have provided care over a longer period (5.93 years) than secondary caregivers (5.50 years). On average, primary caregivers spent more time per day (3.6 hours) providing care to their care recipients than secondary caregivers (2.13 hours). In addition, both primary and secondary caregivers provided personal, financial, emotional and health-related care to their care recipients (Table 2).

Regarding care recipients, their mean age was 77.5 years (Table 3). Most care recipients were females (85%) and had formal education (55%). Also, most care recipients were unemployed (65%), belonged to the Ga-Dangme (90%), Christians (90%), and widowed (65%).

The mean number of children of care recipients was 3.75, and most care recipients were from James Town (75%). In terms of health conditions, most care recipients were living with non-communicable diseases such as hypertension, diabetes, and arthritis. In addition, care recipients receive income each month from their significant others. Remittances from children (100%) and other relatives (50%) and donations from friends (55%) were the primary sources of income for care recipients.

**Table 1. Characteristics of family caregivers.**

| | Primary caregivers (n = 15) | | Secondary caregivers (n = 16) | |
|---|---|---|---|---|
| | Frequency | Percent | Frequency | Percent |
| **Age** | | | | |
| Range (years) | 25–76 | | 21–70 | |
| Mean (years) | 44.9 | | 45.9 | |
| Standard deviation | 16.5 | | 17.4 | |
| **Sex** | | | | |
| Male | 4 | 26.7 | 1 | 6.3 |
| Female | 11 | 73.3 | 15 | 93.7 |
| **Ethnicity** | | | | |
| Ga-Dangme | 13 | 86.7 | 11 | 68.7 |
| Akan | 2 | 13.3 | 2 | 12.5 |
| Other | 0 | 0.0 | 3 | 18.8 |
| **Education** | | | | |
| No education | 1 | 6.7 | 0 | 0.0 |
| Formal education | 14 | 93.3 | 16 | 100.0 |
| **Employment** | | | | |
| Unemployed | 3 | 20.0 | 4 | 25.0 |
| Employed | 12 | 80.0 | 12 | 75.0 |
| **Place of Work*** | | | | |
| At home | 2 | 16.7 | 4 | 33.3 |
| Away from home, nearby | 6 | 50.0 | 6 | 50.0 |
| Away from home, far away | 4 | 33.3 | 2 | 16.7 |
| **Religion** | | | | |
| Christian | 14 | 93.3 | 13 | 81.2 |
| Muslim | 1 | 6.7 | 3 | 18.8 |
| **Marital Status** | | | | |
| Never married | 4 | 26.7 | 7 | 43.8 |
| Currently married | 7 | 46.6 | 2 | 12.4 |
| Formerly married | 4 | 26.7 | 7 | 43.8 |
| **Locality** | | | | |
| James Town | 11 | 73.3 | 11 | 68.7 |
| Ussher Town | 4 | 26.7 | 5 | 31.3 |
| **Relationship with care recipient** | | | | |
| Spouse | 1 | 6.7 | 0 | 0.0 |
| Son/daughter | 9 | 60.0 | 6 | 37.5 |
| Other relative | 5 | 33.3 | 8 | 50.0 |
| Friend | 0 | 0.0 | 2 | 12.5 |
| **Number of years providing care** | | | | |
| Range (years) | 1–15 | | 1–16 | |
| Mean (years) | 5.93 | | 5.50 | |
| Standard deviation | 5.3 | | 4.3 | |
| **Average time spent per day providing care to care recipients** | | | | |
| Range (hours) | 1–6 | | 1–4 | |
| Mean (hours) | 3.60 | | 2.13 | |
| Standard deviation | 1.8 | | 1.2 | |

* Total of place of work = 12 caregivers.

Table 2. Types of care provided by family caregivers.

| Types of care | Caregiving activities performed for care recipients |
|---|---|
| Personal care | Cooking, fetching water for bathing, warming water for bathing, accompanying care recipient to social gatherings, laundering, running errands for care recipient, shopping for groceries, dressing care recipient |
| Financial care | Provision of funds for feeding, payment of hospital bills and medication |
| Emotional care | Chatting with care recipient |
| Health-related care | Accompanying care recipient to the hospital, administering and supervising medication |

## Positive experiences of the primary and secondary caregivers

Several types of rewards were received by the primary and secondary caregivers, and they are described in the succeeding sections. Primary and secondary caregivers received tangible and intangible rewards. Table 4 shows the themes, categories and subcategories.

**Tangible rewards.** *Gifts*. Gifts emerged as a tangible reward for caregiving to older persons. A total of 21 primary and secondary caregivers received gifts as rewards for the provision of care to their care recipients. They received three types of gifts namely: (1) cash; (2) in-kind; and (3) real estate.

Care recipients sometimes financially supported their caregivers when they were financially constrained. Some of them assisted their caregivers to pay their children's school fees. Relatives of the care recipients and churches of both care recipients and caregivers also provided cash to family caregivers to motivate them to continue their care provision. A total of 16 caregivers who received financial gifts were females. The ensuing quotes highlight these financial gifts caregivers received.

> *"Due to the care I provide to my mum, the church we attend sometimes gives me money to motivate me to continue to provide care to her."* (R11)

> *"Sometimes when I am 'broke', my mum supports me financially."* (R31)

In addition, the care recipients, relatives, and friends gave other tangible gifts, including bread, biscuits, cooking oil, and rice, to family caregivers. Family caregivers who received in-kind gifts were females. Some primary and secondary caregivers explained that:

> *"I come [to the care recipient] for milk, provisions, bread, and other supplies. When she has food, she gives me."* (R29)

> *"I sometimes get some provisions due to the care I provide for my mum."* (R11)

Family caregivers also received land and free room/accommodation as a reward for caregiving. One primary caregiver mentioned that he received a piece of land from his father in appreciation for his caregiving roles and responsibilities. He narrated that:

> *"He [care recipient] has given me a piece of land in Kokrobite to build a house. He has been pressuring me to build on it but the challenge is money."* (R18)

In addition, a male primary caregiver noted that the care he provided to his grandmother enabled him to get free room/accommodation in their family house, which would have been impossible due to his young age. He explained that:

**Table 3. Characteristics of care recipients.**

| Background characteristics | Elderly care recipients (N = 20) | |
|---|---|---|
| | Frequency | Percentage |
| **Age** | | |
| Range (years) | 64–95 | |
| Mean (years) | 77.5 | |
| Standard deviation | 10.29 | |
| **Sex** | | |
| Male | 3 | 15.0 |
| Female | 17 | 85.0 |
| **Education** | | |
| No education | 9 | 45.0 |
| Formal education | 11 | 55.0 |
| **Employment** | | |
| Unemployed | 13 | 65.0 |
| Employed | 7 | 35.0 |
| **Ethnicity** | | |
| Ga-Dangme | 18 | 90.0 |
| Ewe | 1 | 5.0 |
| Hausa | 1 | 5.0 |
| **Religion** | | |
| Christian | 18 | 90.0 |
| Islam | 2 | 10.0 |
| **Marital status** | | |
| Never married | 3 | 15.0 |
| Living together | 0 | 0.0 |
| Married | 4 | 20.0 |
| Divorced/Separated | 0 | 0.0 |
| Widowed | 13 | 65.0 |
| **Number of children** | | |
| Range (number of children) | 1–10 | |
| Mean (number of children) | 3.75 | |
| Standard deviation | 2.17 | |
| **Locality** | | |
| James Town | 15 | 75.0 |
| Ussher Town | 5 | 25.0 |
| **Health conditions** | | |
| Communicable diseases (e.g., malaria, fever) | 4 | |
| Non-communicable diseases (e.g., arthritis, stroke, hypertension, and diabetes) | 1 | 5.0 |
| **Sources of income per month***| | |
| Remittances from children | 20 | 100.0 |
| Remittances from other relatives | 10 | 50.0 |
| Pension income | 2 | 10.0 |
| Rental of properties (e.g., land/house) | 1 | 5.0 |
| Income from business venture | 7 | 35.0 |
| Social protection programmes (e.g., Livelihood Empowerment Against Poverty) | 1 | 5.0 |
| Donations from friends | 11 | 55.0 |

*Proportion of care recipients who received a source of income.

**Table 4. Themes, categories and subcategories.**

| Themes | Categories | Sub-categories |
|---|---|---|
| Tangible rewards derived from providing care to their care recipients | Gifts received from care recipients, relatives, friends, and church | Cash gifts received from care recipients, relatives and church |
| | | In-kind gifts received from care recipients, relatives, and friends |
| | | Real estate received from care recipient |
| Intangible rewards derived from providing care to their care recipients | Blessings from God due to providing care | |
| | Skills acquisition from providing care | Skills in providing care to older persons |
| | | Learnt to perform household chores |
| | Enhanced personal attributes of family caregiver | Appreciation of caregiving efforts |
| | | Feeling valued due to privileges caregiving has bestowed on family caregiver |
| | | Learnt humility and patience due to providing care |
| | | Learnt to be compassionate due to providing care |
| | Family cohesion among family due to providing care | |
| | Health consciousness of conditions affecting care recipient | |

*"I have a room [in our family house] and it is because of the care I provide to her."* (R9)

**Intangible rewards.** Five emerging themes from intangible rewards were stated: (1) blessings; (2) skills acquisition; (3) enhanced personal attributes; (4) family cohesion; and (5) health consciousness.

*Blessings.* Seven caregivers (2 males and 5 females) stated that they received blessings from God due to the care they provide to their care recipients and it has consequently opened up opportunities for them. A female primary caregiver explained that:

*"Since I started providing care for my mum, I have never lacked anything. The Good Lord blesses me and sees to it that we never lack anything that we need."* (R5)

The blessings received from their caregiving manifested in various forms. A secondary caregiver explained that he was saved from an accident due to the financial care he provided his care recipients. He narrated the following:

*"All the time, I am being delivered. Like the other time, a tragedy occurred, a serious one. The shop that you see over there [participant pointed to a shop], I hope you have seen the violet shop? We just rebuilt it. It collapsed and I was inside it. Two vehicles crushed and entered it. It even hit me but by the Grace of God, nothing happened to me. So you see, the good things that I do, the good that I use to do to people, you see that the blessing is following me".* (R4)

*Skills acquisition*. The primary and secondary caregivers described their skill acquisition in two areas: (1) skills in providing care; and (2) learnt to perform household chores.

*Skills in providing care*. Four caregivers who provided physical care (such as dressing up the care recipient, fetching water for bathing, cooking, and cleaning care recipients when they soil themselves) reported that they had acquired skills in caring for older persons. These caregivers narrated they had no prior experience in providing care to older persons before they took on their current caregiving roles and responsibilities. All the caregivers who acquired skills in caring for older persons were females. A female primary caregiver who cooks and dresses her grandmother stated the following:

> *"It has helped me to know how to take care of older persons because at first, I didn't know how to do it."* (R12)

*Learnt to perform household chores*. Two family caregivers (1 primary and 1 secondary) narrated that the provision of care to their care recipients has helped them to learn how to perform household chores such as cooking. A secondary caregiver said:

> *"It has taught me how to cook and do household chores."* (R3)

*Enhanced personal attributes*. The primary and secondary caregivers were of the view that caregiving to older persons enhanced their attributes. They expressed enhanced personal attributes in four ways: (1) appreciation; (2) feeling valued; (3) learnt humility and patience; and (4) learnt to be compassionate.

*Appreciation*. Four family caregivers (3 primary and 1 secondary) explained that they felt appreciated as relatives and non-relatives recognised their caregiving efforts that had improved the general well-being of their care recipients. All the caregivers who felt appreciated were females. A primary caregiver commented that:

> *"People even tell me that I have done well by taking care good care of my mum. During the Easter celebration, we went to church and when people saw her, they gave me a handshake that I have done well by taking good care of my mum. When they do that, I feel appreciated."* (R5)

*Feeling valued*. Family caregivers sometimes accompany their care recipients to family meetings. In Ga Mashie and other parts of Ghana, attending and participating in family meetings are often reserve for older persons. However, two female caregivers had the privilege to attend family meetings with their care recipients and this made them feel valued.

> *". . . I feel I am important because I get to sit at places where I would not sit if I was alone. I get to sit at the high table when I accompany her to family meetings. If it was not because of her, I will not be sitting there because my age doesn't qualify me."* (R6)

> *"Because of the care I provide to my care recipient, I can attend family meetings. If not for her, my age would have disqualified me from attending such meetings."* (R24)

*Learnt humility and patience*. Three caregivers (2 primary and 1 secondary) stated that providing care to older persons helped them to cultivate humility and patience. All these caregivers were females. One secondary caregiver narrated that:

> *"It has taught me to be humble and patient in whatever I do."* (R17)

*Learnt to be compassionate.* According to a female secondary caregiver, providing care to her father has helped her to cultivate compassion towards the needy and assisting them when the means are available. She reported that:

*"The help I give to him has helped me learn a lot of things. Some of the things that I have learned are that if someone is in need and you are capable you should help him or her."* (R2)

*Family cohesion.* One female primary caregiver explained that before assuming her caregiving roles and responsibilities, there used to be quarrels within her family, however, provision of care to her mother has bonded the family.

*"There used to be some quarrels in the family but since I started providing care to my mum it has brought the family together."* (R5)

*Health consciousness.* The caregivers who provided health-related care often accompanied their care recipients to hospitals/clinics for health checkups, and this provided them with the opportunity to be educated on health conditions affecting their care recipients. One primary caregiver who cared for her mother diagnosed with hypertension and diabetes explained that access to health education on these conditions made her cautious of her diet and took the necessary precautionary measures to reduce her risk of developing hypertension and diabetes. She noted:

*"Health-wise, I have learned a lot of things that affect the family that can equally affect me so I am taking precautions now and then to make sure I don't fall a victim. So I check my diet."* (R26)

## Discussion

This study explored the positive experiences of primary and secondary caregivers in resource-limited urban settings in Accra, the capital of Ghana. It sought to contribute to researchers' and policymakers' understanding of the nuances of caregiving experiences and why caregivers continue to provide care to older persons despite their negative experiences, which are often reported in the family caregiving literature. Although caregivers may experience some negative outcomes from caregiving, they derive benefits (both tangible and intangible) which motivate them to continue their challenging caregiving roles and responsibilities.

The findings from the study showed that caregivers received gifts from their care recipients, relatives, non-relatives, and religious institutions (such as the church) as a token of appreciation for their caregiving roles and responsibilities. In the Ghanaian setting, the older persons are often given food items, especially bread, when they are visited by their relatives and non-relatives. Caregivers of older persons tend to benefit from these food items. In addition, some older persons occasionally support their caregivers financially. This finding corroborates previous studies in the United States, which indicated that caregivers received financial support from their care recipients [47, 48].

This study found that the family caregivers also received real estate as a reward for the care they provide to their care recipients, namely: land and free room/accommodation. This finding is significant because it is absent in the family caregiving literature, especially in Ghana and the rest of sub-Saharan Africa.

The caregivers believed that providing care to older persons brought blessings to their lives. This belief may be linked to their religiosity and/or spirituality. Previous studies in the United

States also found that caregivers perceived caregiving roles and responsibilities as a blessing [49, 50].

In addition, the findings showed that caregiving to older persons provided family caregivers with the opportunity to learn skills such as performing household chores (for example, cooking and cleaning) and caring for older persons. All the caregivers who reported that they learnt skills through caregiving had no prior caregiving experience. This finding corroborates Fruhauf et al.'s [48] study in the United States which found that family caregiving helped caregivers to acquire the requisite skills to provide care to their care recipients.

Also, the study found that a caregiver became health conscious due to her caregiving roles and responsibilities. Through her interaction with health professionals, she became knowledgeable about hereditary conditions and consequently, became mindful of her diet and thereby reduced her risk of developing those conditions. This finding corroborates Thornton and Hopp's [20] study in the United States, which indicated that daughters became knowledgeable about their parents' health conditions which could help prevent or reduce their risk of developing those conditions.

The findings further showed that caregivers felt appreciated and valued while cultivating humility and patience, and compassion towards the impoverished. The efforts of caregivers to improve the general well-being of their care recipients were recognised by others, and this made the caregivers feel appreciated and proud in their caregiving roles. This result supports previous studies in Hong Kong, Portugal, and the United States, where family caregivers reported that they appreciated the recognition of their caregiving efforts by their care recipients and others [20, 51–53]. Also, previous studies in Hong Kong, Singapore, and the United States found that providing care to care recipients helped family caregivers to develop patience or become more patient [48, 52, 54–56].

Family caregivers also reported that caregiving to their care recipients had strengthened their family bond, especially among siblings Since family members shared their caregiving roles and responsibilities among themselves, they collaborated to enhance their care recipient's well-being, which strengthened the bond among them. This finding supports previous studies that found that family caregiving enhanced relationships among caregivers, care recipients, and other relatives [20, 55–59].

Concerning Kramer's model of caregiver adaptation adopted for this study, the findings showed that caregiver characteristics (such as age and gender) are linked to some positive outcomes (gifts and skills acquisition). With respect to gifts, the findings showed some gender dimensions. For example, only female caregivers received gifts in cash and in-kind, while only male caregivers received gifts in the form of land and access to room/accommodation. Male caregivers often provide financial care [60–62], and which explain why only female caregivers received gifts in cash and in-kind. This finding suggests that interventions to promote positive caregiving outcomes should consider the gender dimensions of family caregiving and the associated rewards.

Regarding age, all family caregivers who learned skills through family caregiving were young and middle-aged adults (25–45 years). This finding suggests that skills acquisition during caregiving may vary by age. Therefore, it is important that specific interventions to promote positive outcomes of caregiving focus on the age of caregivers.

In addition, care recipient characteristics (sources of income per month) influenced some positive outcomes. For instance, the income care recipients received each month from their significant others enabled them to assist their caregivers financially.

The findings of this study have some implications. First, this study found that family caregivers derived tangible and intangible rewards when caring for their care recipients. Such rewards could reduce the burden and burnout of the caregivers. Studies have linked positive

caregiving experiences with a lower caregiving burden [63, 64]. Therefore, policymakers and social workers should consider family caregivers' positive caregiving experiences when designing appropriate interventions to mitigate caregivers' burden and burnout. Second, this study found that younger family caregivers acquired skills in providing care and learnt to perform household chores by providing care to their care recipients. This insight could be incorporated in advocacy for family caregiving to older persons to encourage younger persons to assume family caregiving roles and responsibilities.

## Limitation

This study has two main limitations. First, researchers are using phenomenological design are expected to suspend their views and ideas about the phenomenon they are studying. However, it is impossible to wholly suspend one's views and ideas about a phenomenon of interest. Second, phenomenological design focuses on describing the experiences of individuals than an explanation of those experiences. Despite these limitations, this study has provided comprehensive information on the positive experiences of primary and secondary caregivers and thus contributes to the literature on this neglected aspect of caregiving in Ghana and the rest of sub-Saharan Africa.

## Conclusions

This study demonstrated that family caregivers derive positive experiences from providing care to older persons in their communities. For example, family caregivers received gifts, blessings, acquired skills, and became health conscious. In addition, caregiving enhanced family bonds and personal traits of caregivers. Positive caregiving experiences can mitigate caregivers' burden and burnout. Therefore, policymakers and social workers should design interventions (such as counselling and emotional support) that will enhance the positive experiences of family caregivers. Also, this study found that tangible rewards varied by gender and age. Hence, policymakers and social workers should consider family caregivers' gender and age dynamics in designing interventions to enhance the positive experiences of family caregivers.

## Supporting information

**S1 File. Interview guide for qualitative study.**
(DOCX)

**S1 Appendix. Themes and their sample quotes for rewards family caregiver derived from providing care to their care recipients.**
(DOCX)

## Acknowledgments

The authors would like to thank all participants and research assistants for making this study possible.

## Author Contributions

**Conceptualization:** Frank Kyei-Arthur, Samuel Nii Ardey Codjoe, Delali Margaret Badasu.

**Formal analysis:** Frank Kyei-Arthur.

**Investigation:** Frank Kyei-Arthur.

**Methodology:** Frank Kyei-Arthur.

**Project administration:** Frank Kyei-Arthur.

**Supervision:** Frank Kyei-Arthur, Samuel Nii Ardey Codjoe, Delali Margaret Badasu.

**Validation:** Frank Kyei-Arthur, Samuel Nii Ardey Codjoe, Delali Margaret Badasu.

**Writing – original draft:** Frank Kyei-Arthur.

**Writing – review & editing:** Frank Kyei-Arthur, Samuel Nii Ardey Codjoe, Delali Margaret Badasu.

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
