## [Decision Letter · Decision Letter 0]

24 Feb 2022

PONE-D-21-29801Exploring positive experiences of primary and secondary caregivers of older persons in resource-poor urban settings in Accra, GhanaPLOS ONE

Dear Dr. Kyei-Arthur,

Thank you for submitting your manuscript to PLOS ONE. After careful consideration, we feel that it has merit but does not fully meet PLOS ONE’s publication criteria as it currently stands. Therefore, we invite you to submit a revised version of the manuscript that addresses the points raised during the review process.

Kindly go through the review comments carefully and revise your manuscript. 

We look forward to receiving your revised manuscript.

Kind regards,

Vijayaprasad Gopichandran

Academic Editor

PLOS ONE

" ext-link-type="uri" xlink:type="simple">https://journals.plos.org/plosone/s/fileid=ba62/PLOSOne_formatting_sample_title_authors_affiliations.pdf".

2. We note that you have referenced (Kyei-Arthur F. Family caregiving: Experiences of caregivers and their elderly care recipeints in urban poor communities in Accra, Ghana [Unpublished doctoral thesis]: University of Ghana; 2017.) which has currently not yet been accepted for publication. Please remove this from your References and amend this to state in the body of your manuscript: (ie “Bewick et al. [Unpublished]”) as detailed online in our guide for authors

Reviewers' comments:

Reviewer's Responses to Questions

**Comments to the Author**

1. Is the manuscript technically sound, and do the data support the conclusions?

Reviewer #2: Partly

Reviewer #3: Partly

2. Has the statistical analysis been performed appropriately and rigorously? 

Reviewer #2: N/A

Reviewer #3: Yes

3. Have the authors made all data underlying the findings in their manuscript fully available?

Reviewer #2: Yes

Reviewer #3: Yes

4. Is the manuscript presented in an intelligible fashion and written in standard English?

Reviewer #2: Yes

Reviewer #3: Yes

5. Review Comments to the Author

Reviewer #2: This is an interesting topic of research and the purpose is clearly stated and important. Overall, the manuscript is prepared logically and the data collection seem detailed. There are a few issues the authors need to address. One is the type of this study this falls into. The manuscript is presented in light of qualitative study but no description of that is provided. Authors need to provide that clearly and what type of qualitative study. Secondly, the authors repeatedly used the terms primary and secondary caregivers but have not provided any operational definitions of those terms rather than mentioned them in passing. The tables in the manuscript need to be formatted in a better way like APA style table. The entire paper needs to be edited and formatted too. A couple of minor grammatical errors were made and the authors can correct that using grammerly.

The conclusions need to clearly presented

Reviewer #3: Thanks to the authors for submitting this manuscript which explored positive experiences of caregiving in resource-poor urban settings in Accra, Ghana. There are some major methodological issues that need to be addressed, and the implications of the findings need to be clearly stated to demonstrate the usefulness of this piece of work.

Abstract

• State the research methodology used

• Review the conclusion and make it applicable to the findings of the study. The findings demonstrated that there were intangible and tangible rewards. What is/are the implication(s) of the findings of the study?

Introduction

Page 3, lines 60-64. This section needs to be developed further. The authors made reference to processes influencing the role of the family. Provide information on the nature of influences.

Page 4 – While the author has noted that positive experiences have been reported in the literature, they should be described in the introduction section, rather than just including citations. The majority of readers will not readily go to the reference list and try to search for the articles to identify the positive experiences.

Conceptual Framework

Page 5 – refer to the author by name, instead using the terms she/her.

Materials and Methods

What qualitative method was used in this study?

Page 6 – “family caregivers were purposely selected” – what criteria was used for this selection? This should be stated.

Page 7

The recruitment strategy is unclear

• Line 133 – “The first author and research assistants were able to track 30 out of the 57 family caregivers” – how was this done? The details need to be provided

• Line 136 – Why were care recipients interviewed, if the researcher had ‘tracked’ the caregivers?

The exclusion criteria should be explicitly stated.

What was the aim of the component of the study related to care recipients?

What information was obtained from the care recipients (reference made to 20 in-depth interviews that were done)?

The interview guide captured questions on motivation for providing care, caregiving experiences, coping strategies, perception and other issues. The aim of the study was to capture positive experiences. Some of the information from the interview guide is relevant and appropriate. If the authors indicate in the methods section that coping strategies, for example, was explored, then the finding related to this should also be presented.

Correct the sentence “ethical clearance for the study was approved…” to “ethical clearance for the study was received…”

Did the care recipients provide informed consent?

Did the conceptual framework guide the development of the interview guide?

Review the section on data analysis and remove duplicated information.

When was the qualitative aspect of the study conducted? Was it also done in 2016 when the third wave of the UPHS was conducted? It is not clear.

Operationally define the following:

• Family caregiver

• Primary caregiver

• Secondary caregiver

Results

It may be more meaningful to report the percentages in the narrative rather than the numbers (n).

Review the numbers presented to ensure that they match up to what is in the table. For example, it was indicated that the majority of primary caregivers (n=15) had formal education. In the table, only 14 persons had formal education. Please also noted that formal education is not a term used in the table. Accordingly, the terms in the table should appropriately match what is in the narrative aspect of the results.

Table 1:

• Please report the standard deviation for the mean age

• Add the unit of measurement for age range

• The “n=12” in the row with label “place of work” needs to be reconsidered – why was this done? Were data elements missing?

Adding the label n=12 is confusing. My assumption is that subsequent variables in the table are related to that label (n=12). This is not the case, as the variable ‘religion’, for example, n=15 and n=16 for primary and secondary caregivers respectively would not apply to that label. You could consider using footnotes to reduce confusion.

• Why is the source of the socio-demographic characteristics from the Informal Caregiving Study, 2016? Again, when was the qualitative aspect conducted. Would the characteristics have changed if the qualitative aspect of the study conducted at a later date?

No results have been provided for the care recipient. Please provide an explanation for this, since in-depth interviews were conducted.

Please add details about caregiving experiences of the caregiver, to provide some context to the positive experiences that were reported (for example, information on the number of years providing care, number of hours worked, and the type of care provided).

The legend for Table 2 should be expanded to say what the themes, subthemes and sub-categories relate to.

Discussion

Line 333 – What evidence do you have to conclusively say that the knowledge gap has been filled by your study?

In general, the discussion highlights the findings of the study and predominantly compares the findings with other studies. While this is appropriate and useful, the implications of the findings are not discussed. It therefore begs the question about the usefulness of the study. There is need to strengthen this aspect of the paper.

The conceptual framework is not adequately/appropriately used to discuss the finding of the study. This needs to be addressed.

Limitations

The limitations should be reviewed. The research was not cross-sectional. This should be corrected. It is also well established that the intent of qualitative studies is not to generalize findings. The authors should examine the methodology and identify the limitations of the study.

Conclusion

The conclusion needs to be specific to the findings of the study. What are the implications of the positive experiences identified?

6. PLOS authors have the option to publish the peer review history of their article (what does this mean?). If published, this will include your full peer review and any attached files.

Reviewer #2: No

Reviewer #3: No

---

## [Author Response · Author response to Decision Letter 0]

16 Mar 2022

Responses to academic editor and reviewers’ comments

Academic Editor 

Response: We thank the academic editor for the comment and suggestion. We have revised the entire manuscript to ensure it meets PLOS ONE requirements. 

2. We note that you have referenced (Kyei-Arthur F. Family caregiving: Experiences of caregivers and their elderly care recipeints in urban poor communities in Accra, Ghana [Unpublished doctoral thesis]: University of Ghana; 2017.) which has currently not yet been accepted for publication. Please remove this from your References and amend this to state in the body of your manuscript: (ie “Bewick et al. [Unpublished]”) as detailed online in our guide for authors

Response: We have removed the unpublished manuscript from the references and we have replaced the in-text reference with (Kyei-Arthur [Unpublished]) in the revised manuscript. See page 5-6, lines 112-114.

“This manuscript is derived from a broader thesis that collected data on characteristics of family caregivers and their care recipients, and caregivers’ motivations for providing care to their care recipients (Kyei-Arthur [Unpublished]).”

3. Please include captions for your Supporting Information files at the end of your manuscript, and update any in-text citations to match accordingly.

Response: We have included a caption for our Supporting Information at the end of our manuscript. See page 34, lines 632-636.

Reviewer 2

1. One is the type of this study this falls into. The manuscript is presented in light of qualitative study but no description of that is provided. Authors need to provide that clearly and what type of qualitative study.

Response: We have specified that the study used a phenomenological design in the revised manuscript. See page 2, lines 29-30 and page 6, lines 126-129.

2. The authors repeatedly used the terms primary and secondary caregivers but have not provided any operational definitions of those terms rather than mentioned them in passing.

Response: We thank the reviewer for the comment. We have defined who primary and secondary caregivers are in the data collection section in the manuscript. Also, we defined who family caregivers are. See page 8, lines 159-163.

3. The tables in the manuscript need to be formatted in a better way like APA style table.

Response: We thank the reviewer for the suggestion. Unfortunately, PLOS ONE does not used APA style for its table formatting. We have formatted all tables in the revised manuscript to meet the table format of PLOS ONE.

4. The entire paper needs to be edited and formatted too.

Response: We have edited and formatted the revised manuscript.

5. A couple of minor grammatical errors were made and the authors can correct that using grammerly.

Response: We have edited the revised manuscript to address the grammatical errors. 

6. The conclusions need to clearly presented.

Response: We have made the conclusion section specific in the revised manuscript. See page 24-25, lines 454-462.

Reviewer 3

1. Abstract - State the research methodology used.

Response: We have specified that the study used phenomenology design in the revised manuscript. See page 2, line 29-30. 

2. Abstract - Review the conclusion and make it applicable to the findings of the study. The findings demonstrated that there were intangible and tangible rewards. What is/are the implication(s) of the findings of the study?

Response: We have included the implications of the findings of the study in the abstract in the revised manuscript. See page 2, lines 35-38.

3. Introduction - Page 3, lines 60-64. This section needs to be developed further. The authors made reference to processes influencing the role of the family. Provide information on the nature of influences.

Response: We have provided details on how modernisation processes influence care for older persons in the revised manuscript. See page 3, lines 64-67.

“For instance, the increased participation of women in the labour market affects the availability of women to provide personal care to older persons [15]. In addition, research has established that financial constraints limit the financial care provided to parents [16].”

4. Introduction - Page 4 – While the author has noted that positive experiences have been reported in the literature, they should be described in the introduction section, rather than just including citations. The majority of readers will not readily go to the reference list and try to search for the articles to identify the positive experiences.

Response: We thank the reviewer for the comment. We have indicated some of the positive experiences of family caregivers in the introduction section of the revised manuscript. See page 4, lines 71-74.

5. Revise the sentence “Between 1990 and 2021, their proportion of the total global population increased from 6 percent to 10 percent, and it is estimated to reach 16 percent in 2050 [1, 2].” to “Between 1990 and 2021, the proportion of the population of older people to total global population increased from 6 percent to 10 percent, and it is estimated to reach 16 percent in 2050 [1, 2]”.

Response: I have revised the sentence in the revised manuscript. See page 3, lines 48-50.

6. Replace “their” with “the” in the sentence “In Ghana, their proportion has increased from 3 percent in 1960 to 4 percent in 2021 [2, 3]”.

Response: We have replaced “their” with “the” in the sentence in the revised manuscript. Please see page 3, line 50.

7. Conceptual Framework - Page 5 – refer to the author by name, instead using the terms she/her.

Response: We have used the name of the author “Kramer” to replace “her” in the revised manuscript. Please see page 5, line 100.

8. Replace “article” with “manuscript” in the manuscript.

Response: We have replaced “article” with “manuscript” in the revised manuscript. Please see page 5, line 112 and page 6, line 119.

9. Materials and Methods - What qualitative method was used in this study?

Response: We have specified that the study used phenomenology design in the revised manuscript. See page 6, lines 126-129.

In addition, we have revised the heading “Study setting” to “Design and study setting” in the revised manuscript. See page 6, line 125.

10. Revise the heading “Materials and Methods” to “Methods”.

Response: We have revised the heading “Materials and Methods” to “Methods” in the revised manuscript. Please see page 6, line 124.

11. I think using communities will be more appropriate than towns.

Response: We have replaced “towns” with “communities” in the revised manuscript. Please see page 6, line 131.

12. Replace “poor” with “limited” in the manuscript.

Response: We have replaced “poor” with “limited” in the revised manuscript. See page 1, line 5; page 2, line 29 and 39; page 5, line 93; page 6, line 133; and page 20, line 367.

13. Page 6 – “family caregivers were purposely selected” – what criteria was used for this selection? This should be stated.

Response: In the first stage of the data collection, family caregivers were not purposely selected. We apologise for the misinformation. We have revised the entire data collection section in the revised manuscript to make it clearer. See page 7-8, lines 140-176.

14. Page 7

The recruitment strategy is unclear

• Line 133 – “The first author and research assistants were able to track 30 out of the 57 family caregivers” – how was this done? The details need to be provided

Response: We have provided more details on how the research team was able to contact the 30 family caregivers in the revised manuscript. See page 7, lines 152-158.

“In total, the first author and trained research assistants were able to contact 30 out of the 57 family caregivers using the structure and household identifiers (e.g., enumeration area number, structure number of household, name of the head of household, and contact of the head of household) and family caregiver identifiers (e.g., name, and contact numbers). In addition, care recipients assisted the research team in reaching out to all persons who provided care to them since the study was interested in primary and secondary caregivers.”

15. • Line 136 – Why were care recipients interviewed, if the researcher had ‘tracked’ the caregivers?

Response: We thank the reviewer for the comment. Care recipients were interviewed because we wanted to identify all persons (primary and secondary caregivers) provided care to them. Although interviewing family caregivers we tracked could helped us to identify primary and secondary caregivers, we were of the opinion that interviewing the care recipients first and interviewing primary and secondary caregivers they identified was the best way to go about it since caregiving roles and responsibilities changes over time. It is the caregiving roles and responsibilities which are used to determine who is the primary or secondary caregiver.

16. The exclusion criteria should be explicitly stated.

Response: We have explicitly reported the exclusion criteria in the revised manuscript. See page 8, lines 167-170. 

17. What was the aim of the component of the study related to care recipients?

Response: The aim of the study related to care recipient was to explore the perspectives of care recipients on the care they received from their caregivers. This was part of the larger project. However, it is not focus of this manuscript. This manuscript focused on the positive experiences of family caregivers. In the methods section of the revised manuscript, we have deleted most of the details about care recipients since it is not the focus of this manuscript. It is worth noting that we described the characteristics of care recipients in the revised manuscript. See page 10, lines 215-223, and Table 3 on page 12-13, line 231-232.

In addition, we have revised the subheading “socio-demographic characteristics of family caregivers” to “socio-demographic characteristics of family caregivers and their care recipients” in the revised manuscript. See page 9, lines 198-199.

18. What information was obtained from the care recipients (reference made to 20 in-depth interviews that were done)?

Response: Care recipients were asked their perspectives on the care they received from their caregivers. We have described the characteristics of care recipients in the revised manuscript. See page 10, lines 215-223, and Table 3 on page 12-13, line 231-232.

19. The interview guide captured questions on motivation for providing care, caregiving experiences, coping strategies, perception and other issues. The aim of the study was to capture positive experiences. Some of the information from the interview guide is relevant and appropriate. If the authors indicate in the methods section that coping strategies, for example, was explored, then the finding related to this should also be presented.

Response: We thank the reviewer for the observation. Although information on coping strategies was collected using the interview guide, we have already published an earlier manuscript which focused on family caregivers’ challenges and their coping strategies. Consequently, we decided not to present such data since it has already been published elsewhere and focused on family caregivers’ positive caregiving experiences which has not been published.

20. Correct the sentence “ethical clearance for the study was approved…” to “ethical clearance for the study was received…”

Response: We have corrected the sentence “ethical clearance for the study was approved…” to “ethical clearance for the study was received…” in the revised manuscript. See page 8, lines 170-171. 

21. Did the care recipients provide informed consent?

Response: Care recipients gave their informed consent as well as family caregivers. All care recipients had no cognitive impairment so they were able to provide informed consent before they were interviewed. 

22. Did the conceptual framework guide the development of the interview guide?

Response: The conceptual framework partly guided the development of the interview guide. The review of literature and pilot of the interview guide also influenced the development of the interview guide.

23. This data was collected over five years ago. Don’t you think the findings being reported here may be obsolete? Or don’t you think things may have changed significantly between 2016 and 2022?

Response: We appreciate the concern of the reviewer. We agree that there could be significant changes in the experiences of caregivers between 2016 and 2022. Despite the potential changes in caregivers’ experiences, it is worth sharing their experiences as of 2016 since their experiences could help in the design of interventions to enhance family caregivers’ positive caregiving experiences in Ghana.

24. Review the section on data analysis and remove duplicated information.

Response: We have revised the data analysis section in the revised manuscript. See page 8-9, lines 179-195.

25. You need to remove yourself from the data and allow it to assume neutrality of how it was handled.

Response: We thank the reviewer for the suggestion. We have removed ourselves from the data analysis and revised the data analysis section in the revised manuscript. See page 8-9, lines 179-195.

26. When was the qualitative aspect of the study conducted? Was it also done in 2016 when the third wave of the UPHS was conducted? It is not clear.

Response: The Urban Health and Poverty Survey was conducted in September 2013 and the qualitative study was conducted in November 2016. We have revised the data collection section in the revised manuscript to make things clearer. See pages 7-8, lines 140-176. 

27. Operationally define the following:

• Family caregiver

• Primary caregiver

• Secondary caregiver

Response: We have defined family caregiver, primary caregiver and secondary caregiver in the data collection section in the revised manuscript. See pages 8, lines 159-163. 

28. It is important that the authors provide definitions of primary and secondary caregivers for this research. I know you mentioned above who they are but it will help to clearly answer that question before readers out ask it when you are not there to explain. Provide a paragraph under methods to explain who primary and secondary caregivers are for the sake of your study.

Response: We have defined primary caregiver and secondary caregiver in the data collection section in the revised manuscript. We also defined family caregiver in the revised manuscript. See pages 8, lines 159-163. 

29. Results - It may be more meaningful to report the percentages in the narrative rather than the numbers (n).

Response: We have used percentages in the narratives as suggested by the reviewer in the revised manuscript. See page 9-10, lines 200-223.

30. Review the numbers presented to ensure that they match up to what is in the table. For example, it was indicated that the majority of primary caregivers (n=15) had formal education. In the table, only 14 persons had formal education. Please also noted that formal education is not a term used in the table. Accordingly, the terms in the table should appropriately match what is in the narrative aspect of the results.

Response: We thank the reviewer for the observation. We have reconciled the figures and terms used in the Tables and narrative in the revised manuscript. See page 9-13, lines 200-232. 

31. Table 1 - Please report the standard deviation for the mean age

Response: We have added the standard deviation of the mean age to Table 1 in the revised manuscript. See page 11, line 225.

32. Table 1 - Add the unit of measurement for age range

Response: Age range was measured in years. We have indicated years as the unit of measurement in the revised manuscript. See page 11, line 225.

33. The majority of the primary caregivers (n = 15) had formal education, while all 16 secondary caregivers had formal education. Please, read this phrase and correct accordingly. Which group had formal education?

Response: We have revised the sentence “The majority of the primary caregivers (n = 15) had formal education, while all 16 secondary caregivers had formal education.” To “The majority of the secondary caregivers (100%) had formal education compared to primary caregivers (93.3%).” in the revised manuscript. Please see page 10, line 203-204.

34. Table 1 - The “n=12” in the row with label “place of work” needs to be reconsidered – why was this done? Were data elements missing?

Response: We apologise for using “n=12” in the manuscript. It was used to show that the total for “place of work” is less than the total of the other items/indicators/variables in the table. We have deleted the “n=12” in the revised manuscript and used asterisks on the “place of work” and explained below the Table that the sum/total of the “place of work” is 12 caregivers. See page 11-12, line 225-226.

35. Adding the label n=12 is confusing. My assumption is that subsequent variables in the table are related to that label (n=12). This is not the case, as the variable ‘religion’, for example, n=15 and n=16 for primary and secondary caregivers respectively would not apply to that label. You could consider using footnotes to reduce confusion.

Response: We thank the reviewer for the suggestion. We have deleted the “n=12” in the revised manuscript and used asterisks on the “place of work” and explained below the Table that the sum/total of the “place of work” is 12 caregivers. See page 11-12, line 225-226.

36. Why is the source of the socio-demographic characteristics from the Informal Caregiving Study, 2016?

Response: We are sorry for the misinformation. The “Informal Caregiving Study, 2016” was a name the first author gave to the qualitative study that was conducted. We have deleted the source of data as “Informal Caregiving Study, 2016” in the revised manuscript to avoid any confusion. 

37. Again, when was the qualitative aspect conducted. Would the characteristics have changed if the qualitative aspect of the study conducted at a later date?

Response: The qualitative data was conducted in November 2016. Some characteristics of family caregivers will change if it was conducted at a later date, especially in 2017 and beyond. 

38. No results have been provided for the care recipient. Please provide an explanation for this, since in-depth interviews were conducted.

Response: We have provided information on the characteristics of care recipients in the revised manuscript. However, it is worth noting that this study focused on family caregivers. See page 10, lines 215-223 and Table 3 in pages 12-13, line 231-232.

39. Please add details about caregiving experiences of the caregiver, to provide some context to the positive experiences that were reported (for example, information on the number of years providing care, number of hours worked, and the type of care provided).

Response: We have provided details on the number of years providing care, average time spent providing care to care recipients, and the type of care provided in the revised manuscript in Table 1 and Table 2. See page 10, lines 210-214; Table 1 in pages 11-12, lines 225-226; and Table 2 in page 12, lines 228-229.

40. The legend for Table 2 should be expanded to say what the themes, subthemes and sub-categories relate to.

Response: We have revised the themes, categories and sub-categories in Table 3 and supporting information (S2 Appendix) to make it easier for readers to understand the themes, categories, and sub-categories. See page 14, line 242.

41. Communicating with older persons - You need more than just narrative here to prove the case

Response: The theme on communicating with older persons had only one quote so we have deleted it from results and discussion sections in the revised manuscript.

42. Discussion - Line 333 – What evidence do you have to conclusively say that the knowledge gap has been filled by your study?

Response: We thank the reviewer for the comment. We cannot conclusively prove that our study has filled the knowledge gap. Consequently, we have deleted the sentence “The examination of this aspect of caregiving has been limited or even absent in the literature on caregiving in Ghana and the rest of sub-Saharan Africa. This knowledge gap in Ghana is filled by this study.” in the revised manuscript. 

43. “The examination of this aspect of caregiving has been limited or even absent in the literature on caregiving in Ghana and the rest of sub-Saharan Africa. This knowledge gap in Ghana is filled by this study.” This has already stated in the introduction. 

Response: We have deleted the two sentences in the revised manuscript. 

44. Replace “interviews with family caregivers” and “they” with “study” and “caregivers” respectively on line 334. 

Response: We have replaced “interviews with family caregivers” and “they” with “study” and “caregivers” respectively in the revised manuscript. See page 21, line 373.

45. In general, the discussion highlights the findings of the study and predominantly compares the findings with other studies. While this is appropriate and useful, the implications of the findings are not discussed. It therefore begs the question about the usefulness of the study. There is need to strengthen this aspect of the paper.

Response: We have included information on implications of the findings in the discussion section of the revised manuscript. See page 23-24, lines 432-441. 

46. The conceptual framework is not adequately/appropriately used to discuss the finding of the study. This needs to be addressed.

Response: We thank the reviewer for the comment. The conceptual framework was used in the larger project but this study focuses on some aspects of the framework (i.e. positive caregiving experiences). We used the conceptual framework to describe the positive experiences of family caregivers rather the pathways to positive caregiving. We demonstrated that some caregivers characteristics (age and sex) influence positive caregiving in the manuscript. We have also described how care recipients’ characteristics, specifically their sources of income per month, influence family caregiver positive caregiving experiences in the revised manuscript. See page 23, lines 429-431. 

In addition, we have revised the conceptual framework adopted (Fig. 1) for this study to make it clearer. It is worth noting that caregiver attitudinal variables and efforts, and resources in the conceptual framework were not the focus of this study. 

47. Limitations - The limitations should be reviewed. The research was not cross-sectional. This should be corrected. It is also well established that the intent of qualitative studies is not to generalize findings. The authors should examine the methodology and identify the limitations of the study.

Response: We have revised the limitation section in the revised manuscript. See page 24, lines 444-451. 

48. Conclusion - The conclusion needs to be specific to the findings of the study. What are the implications of the positive experiences identified?

Response: We have revised conclusion section in the revised manuscript. See page 24-25, lines 454-462.

---

## [Editor Report · Decision Letter 1]

18 Mar 2022

Exploring positive experiences of primary and secondary caregivers of older persons in resource-limited urban settings in Accra, Ghana

PONE-D-21-29801R1

Dear Dr. Kyei-Arthur,

We’re pleased to inform you that your manuscript has been judged scientifically suitable for publication and will be formally accepted for publication once it meets all outstanding technical requirements.

Kind regards,

Vijayaprasad Gopichandran

Academic Editor

PLOS ONE
---

## [Editor Report · Acceptance letter]

24 Mar 2022

PONE-D-21-29801R1 

Exploring positive experiences of primary and secondary caregivers of older persons in resource-limited urban settings in Accra, Ghana 

Dear Dr. Kyei-Arthur:

I'm pleased to inform you that your manuscript has been deemed suitable for publication in PLOS ONE. Congratulations! Your manuscript is now with our production department. 

Kind regards, 

on behalf of

Dr. Vijayaprasad Gopichandran 

Academic Editor

PLOS ONE